# Effects of Vaccination Day Routine Activities on Influenza Vaccine Efficacy and Vaccination-Induced Adverse Reaction Incidence: A Cohort Study

**DOI:** 10.3390/vaccines9070753

**Published:** 2021-07-06

**Authors:** Tsuneaki Kenzaka, Shinsuke Yahata, Ken Goda, Ayako Kumabe, Momoka Kamada, Masanobu Okayama

**Affiliations:** 1Division of Community Medicine and Career Development, Kobe University Graduate School of Medicine, Hyogo, Kobe 652-0032, Japan; kenkenpetneed@yahoo.co.jp (K.G.); kumabe-kmm@umin.ac.jp (A.K.); 2Department of Internal Medicine, Hyogo Prefectural Tamba Medical Center, Hyogo, Tamba 669-3495, Japan; m08018mo@jichi.ac.jp; 3Division of Community Medicine and Medical Education, Kobe University Graduate School of Medicine, Hyogo, Kobe 652-0032, Japan; yahata-jci@umin.ac.jp (S.Y.); okayamam@med.kobe-u.ac.jp (M.O.); 4Department of Internal Medicine, Shiso Municipal Hospital, Hyogo, Shiso 671-2576, Japan; 5Department of Internal Medicine, Toyooka Public Hospital, Hyogo, Toyooka 668-8501, Japan

**Keywords:** influenza, vaccination, routine activity, bath, smoking, alcohol consumption, caffeine intake, exercise

## Abstract

We aimed to investigate the effect of vaccination day routine activities on the influenza vaccine efficacy and vaccination-induced adverse reaction incidence. Study participants were workers at three hospitals in the Hyogo Prefecture, Japan, who received the influenza vaccine between October and November, 2018 and 2019. Their data were collected using a questionnaire. The main factors, which were examined for vaccination day routine activities, were personal hygiene (bathing), smoking, alcohol consumption, caffeine intake, and exercise. The main outcome markers included influenza incidence during the winter season and vaccination-induced local or systemic adverse reaction incidence. The risk ratio for the main factors of vaccination day routine activities was calculated against the main outcome markers using the modified Poisson regression. Overall, 3780 people received the influenza vaccination, and 2731 submitted the questionnaire. We found that vaccination day routine activities did not affect the influenza vaccine efficacy, and engaging in strenuous exercise on the vaccination day tended to cause systemic adverse reactions. Moreover, lifestyle-related activities had no impact on the incidence of systemic or local adverse reactions. Therefore, it is advisable to only avoid strenuous physical exercise, and no other lifestyle-related restrictions are necessary on the day of influenza vaccination.

## 1. Introduction

Influenza is an infectious disease that is most prevalent during winter. Japan’s National Institute of Infectious Diseases has reported that out of a population of 126 million, 12 million (9.5%) and 7.28 million (5.7%) people contracted influenza during the winter seasons between 2018 and 2019, and 2019 and 2020, respectively [1]. In 2018, there were 3325 reported deaths due to influenza in Japan [1].

The administration of vaccines against the infection caused by the influenza virus has been a common practice and is the single most effective prevention strategy [2]. Because the majority of the individuals worldwide is expected to remain susceptible to one or more circulating influenza strains at any given time, further understanding of optimal vaccine practices is essential. Inactivated influenza vaccine is prepared by killing influenza viruses and isolating their subvirion or purified surface antigens to create “split” or subunit vaccines [2]. The inactivated influenza vaccine is then introduced into an individual’s immune system through injection. This causes rapid distribution of the antigen into the bloodstream and induces a humoral response [2].

Daily exercise [3] lowers the risk of contracting influenza, whereas smoking [4] and alcohol consumption [5] increase the risk. However, these factors do not alter the efficacy of the influenza vaccine [2]. A survey study on US military members reported that there was no significant difference in the efficacy of influenza vaccine with relation to daily routine activities between the groups of smokers and nonsmokers, those with a potential alcohol problem and those without an alcohol problem, and those reporting a high exercise level and those reporting a minimal exercise level [2]. However, another study reported that daily alcohol consumption increases the risk of developing adverse reactions after influenza vaccination [6].

In Japan, where the use of public bathhouses is part of the culture, people are usually advised to avoid taking a bath or engaging in exercise after receiving a vaccine, including the influenza vaccine. This is because of the risk of localized infection via the injection site while in the bath or the difference in temperature between the changing and bathing rooms, which may cause people to get sick [7]. At public bathhouses, the bathtubs may be used multiple times by many individuals consecutively. Therefore, hygiene of the hot water in the bath may not be as good as in a private home. In fact, vaccination guidelines before 1994 instructed people to avoid bathing and urged people to rest on the vaccination day [7].

In Japan, healthcare professionals often document some details of the vaccinated individuals, including routine activities performed on the day of vaccination, such as personal hygiene (bathing), smoking, alcohol consumption, or exercising, to assess the efficacy and adverse reactions of the influenza vaccine. However, to the best of our knowledge, no reports exist regarding the influence of routine activities performed on the day of vaccination, such as personal hygiene (bathing), smoking, alcohol consumption, or exercising, on the efficacy and adverse reactions of the influenza vaccine. These results could affect lifestyle-related activities on the vaccination day.

Therefore, we aimed to investigate the effect of the abovementioned routine activities performed on the vaccination day on vaccine efficacy and adverse reaction incidence after vaccination.

## 2. Materials and Methods

### 2.1. Study Design

This cohort study was conducted with the approval of the Ethics Committee of Hyogo Prefectural Tamba Medical Center (Approval No. Tan-I No. 1166). Participants provided their written informed consent for study participation and result publication.

### 2.2. Participants

The study participants were all vaccinated employees at three hospitals in Hyogo Prefecture (Hyogo Prefectural Tamba Medical Center, Toyooka Public Hospital, and Shiso Municipal Hospital) who received the influenza vaccine between October and November 2018 or 2019. Mass vaccination had been performed for all employees during this period. In Japan, a mass influenza vaccination program for healthcare professionals is conducted in hospitals every year during the influenza season. All healthcare professionals in the three hospitals who took the influenza vaccine were eligible to participate in our study.

### 2.3. Setting

Each participant was provided with a vaccine screening healthcare questionnaire prior to vaccine administration. This is a standard procedure in Japan, and the standardized questionnaire was analyzed in this study. The questions included participant characteristics, their routine activities on the vaccination day, and the incidence of adverse reactions after vaccination. The questionnaire was collected 10 days after the participants were vaccinated with reference to a previous study [8].

### 2.4. Details of the Questionnaire

Participant characteristics included the following: age, sex, pregnancy status (women only), underlying conditions, body temperature at the time of vaccination, poor physical condition at the time of vaccination, influenza vaccination history, and food/drug allergy history.

Routine activities on the vaccination day included the following: personal hygiene (bath, shower only, washed using a towel or not washed), smoking (nonsmoker, previous smoker, smoker but did not smoke on vaccination day, smoker and smoked on vaccination day), alcohol consumption (consumed alcohol or did not consume alcohol on vaccination day), caffeine intake (consumed caffeine or did not consume caffeine on vaccination day), and exercise with perspiration (exercised or did not exercise on vaccination day).

The incidence of the following local adverse reactions on injection site was determined: redness, swelling, induration, pain, heat sensation, pruritus, and heaviness/lassitude.

The incidence of the following systemic adverse reactions was determined: fever, chills, headache, fatigue, nasal discharge, cough, nausea, diarrhea, difficulty moving the upper limbs, and numbness.

### 2.5. Incidence of Influenza

Influenza was diagnosed using rapid influenza diagnostic tests, which are immunoassays that can identify the presence of influenza A and B viral nucleoprotein antigens. The questionnaires provided prior to vaccine administration, including those analyzed in this study, were sequentially numbered before distribution and managed by the Infection Control Office of each hospital, independent of the study organizer. Each Infection Control Office confirmed the identity of the participants who contracted influenza during the 2018–2019 and 2019–2020 winter seasons among those vaccinated in 2018 and 2019, respectively. Individuals affected by influenza were matched to the numbers on the questionnaires to confirm the incidence of influenza during each season.

Using the abovementioned method, participant characteristics, presence or absence of adverse reactions, and incidence of influenza were blinded from researchers to protect the participants’ privacy.

### 2.6. Definition of Underlying Conditions

Underlying conditions were defined and classified as the presence or absence of the following conditions: cancer, immunological disease/diseases caused by steroid use, renal disease, diabetes, hypertension, hepatic disease, and pulmonary disease.

### 2.7. Data Analysis of Participants’ Background Characteristics

Means of continuous variables and percentiles of categorical variables were calculated to characterize participants.

### 2.8. Main Factors

Routine activities on vaccination day are listed below.

Personal hygiene: control (no washing, wipe only, or shower only) vs. bath.Smoking: control (nonsmoker) vs. previous smoker, smoker but did not smoke on the vaccination day, or smoker and smoked on the vaccination day.Alcohol consumption: control (did not consume alcohol on the vaccination day) vs. consumed alcohol on the vaccination day.Caffeine intake: control (no caffeine) vs. consumed caffeine.Exercise: control (did not exercise on the vaccination day) vs. exercised on the vaccination day.

### 2.9. Main Outcome Markers

Contracted influenza during the season: no/yes.Adverse reaction after vaccinationSome types of adverse reaction: none, local, or systemic adverse reaction.Local adverse reaction: none and some types of local adverse reaction.Systemic adverse reaction: none and some types of systemic adverse reaction.

### 2.10. Statistical Analysis

The risk ratio and 95% confidence interval for the main factors from the routine activities on vaccination day were calculated against the main outcome markers using modified Poisson regression. Analysis was performed based on the following three models.

Model 1Rough analysis.Model 2Adjustment for sex, age, and underlying conditions.Model 3Model 2 + adjustment for routine activities (personal hygiene, smoking, alcohol consumption, caffeine intake, and exercise).

Data were analyzed using Stata MP version 15 (StataCorp, College Station, TX, USA).

## 3. Results

The fractions of the numbers of questionnaires received and people who were vaccinated at Hyogo Prefectural Tamba Medical Center were 470/485 (collection rate of 96.91%) and 656/717 (collection rate of 91.49%) in 2018 and 2019, respectively. The fractions of the numbers of questionnaires received and people who were vaccinated at Toyooka Public Hospital were 505/974 (collection rate of 51.85%) and 622/999 (collection rate of 62.26%) in 2018 and 2019, respectively. The fractions of the numbers of questionnaires received and people who were vaccinated at Shiso Municipal Hospital were 233/291 (collection rate of 80.06%) and 245/314 (collection rate of 78.03%) in 2018 and 2019, respectively. In total, 3780 people received the influenza vaccination, and 2731 submitted the questionnaire (collection rate of 72.24%). Participant characteristics, routine activities on vaccination day, and incidence of local and/or systemic adverse reactions are shown in Table 1 (missing values were excluded from calculations).

The mean age ± standard deviation was 42.7 ± 12.7 years, and women accounted for 76.18% of the participants. The majority of participants had no underlying conditions.

Regarding routine activities on vaccination day, 70.57% of the participants took a bath, 11.02% were previous smokers, 0.41% were smokers but did not smoke, 6.34% were smokers and smoked on that day, 16.97% drank alcohol, 49.83% consumed caffeine, and 4.95% exercised.

The number of people who contracted influenza was 146 (5.35%). During the 2018–2019 and 2019–2020 winter seasons, 78 of 1130 (6.46%) and 68 of 1444 (4.71%) participants contracted influenza, respectively.

Of the vaccinated participants, 81.67% and 19.99% experienced local and systemic adverse reactions, respectively; 82.66% experienced either a local or systemic adverse reaction.

Table 2 shows the effects of vaccination day routine activities on influenza incidence.

From the risk ratio, none of the activities significantly affected the incidence of influenza.

Table 3 shows the effect of vaccination day routine activities on the incidence of adverse reactions.

Table 4 shows the effect of vaccination day routine activities on the incidence of local adverse reactions.

From the risk ratio, none of the activities significantly affected the adverse reactions.

Table 5 shows the effect of vaccination day routine activities on the incidence of systemic adverse reactions.

Exercising significantly increased the risk of developing systemic adverse reactions by 44%, with a risk ratio of 1.44 (95% confidence interval: 1.06–1.97). Frequently observed systemic adverse reactions (Table 1) were fatigue (8.67%), difficulty moving the upper limbs (7.14%), and headache (4.89%).

## 4. Discussion

In the present study, we found that vaccination day routine activities, such as personal hygiene (bathing), smoking, alcohol consumption, caffeine intake, and exercise, did not affect the efficacy of the influenza vaccine. Furthermore, we found that engaging in strenuous exercise on the vaccination day tended to cause systemic adverse reactions. Moreover, apart from exercise, lifestyle-related activities had no impact on the incidence of systemic and local adverse reactions.

According to previous study findings, daily exercise lowers the risk of contracting influenza [2]. This result is possibly attributed to improved viral clearance [9], increased antibody titer levels [10], or the effect of neuroendocrine factors [11]. Conversely, the risk of contracting influenza has been reported to increase with smoking [3] and alcohol consumption [4]. Tobacco components have been suggested to block antiviral pathways and increase susceptibility to influenza through oxidative stress-associated mechanisms [12]. Moreover, smoking decreases the host defense properties of the respiratory mucosa [13]. Furthermore, chronic alcohol consumption may cause an inflammatory environment and alter CD8T cell response, thereby increasing susceptibility to influenza viruses [4]. Moreover, previous studies have reported that daily habitual smoking, alcohol consumption, and regular exercise have no effect on influenza vaccine efficacy [5]. We found that lifestyle factors, such as personal hygiene (bathing), smoking, alcohol consumption, caffeine intake, and exercise, had no effect on the influenza vaccine efficacy. Before 1994, people were advised to avoid bathing and were encouraged to rest on the vaccination day [7]. However, the basis of such instructions hitherto, remains unclear. Our study findings showed, for the first time, that such restrictions were unnecessary prior to receiving the influenza vaccine and that people should continue to perform their normal daily activities on the vaccination day. On the vaccination day, only strenuous exercise that could cause sweating and heavy alcohol consumption should be avoided, as noted in the vaccination guidelines [7], and no other lifestyle-related restrictions are unnecessary.

Although daily alcohol consumption has been reported to increase the risk of developing adverse reactions [6], no special consideration has been given to the mechanism of alcohol consumption. Systemic adverse reactions that were frequently observed included fatigue (8.67%), difficulty moving the upper limbs (7.14%), and headache (4.89%), which may be caused by an increased immune response associated with exercise. It has been reported that an increase in the serum levels of macrophage migration inhibitory factor is involved in systemic adverse reactions after vaccination [14]. We believe that the increase in macrophage migration inhibitory factor due to exercise is associated with the increase in the incidence of systemic adverse reactions [15].

In this study, the incidences of influenza among participants were 6.46% and 4.71% during the 2018–2019 and 2019–2020 winter seasons, respectively. We believe that the findings of our study are important because the national influenza incidence rates for Japan were 9.5% and 5.7% [1] during the 2018–2019 and 2019–2020 winter seasons, respectively, and all participants were healthcare professionals who have sufficient knowledge regarding infection control measures. A study conducted in Australia and the Philippines with healthy adults reported that approximately 50% of participants had some types of adverse reactions after influenza vaccination [16]. Another study conducted in Japan among healthcare professionals reported that 73.9% and 15.8% of the professionals had local and systemic reactions, respectively [8]. In the present study, 81.67% and 19.99% of participants had local and systemic adverse reactions, respectively, which corroborated with previous study findings. Therefore, we believe that the incidence of influenza and adverse reaction incidence used as outcome markers in our study were reasonable.

### Limitations

Adverse reactions were self-reported and not objectively monitored. Although the study participants were all healthcare professionals with a certain level of reliability, the participants’ subjectivity may still be observed. In addition, because it is easy to track a large number of people, this group was targeted. The collection rate of 72.24% in this study was high; in contrast, the collection rate for Japanese questionnaire survey has been reported to be 14–18% [17].

It has been reported that, in the Japanese population, >30% of men and 10% of women smoke [18], 19–29% of men and 2–11% of women drink alcohol [19], and 10–30% of people exercise regularly [20]. In the present study, 6.34%, 16.97%, and 4.95% of participants smoked, consumed alcohol, and exercised, respectively. There was a possibility of selection bias as the participants were medical professionals. Hence, care should be taken when generalizing the results of this study. If the general public is included in the target population, the proportion of individuals smoking and/or consuming alcohol on the day of vaccination will increase, and the proportion of individuals exercising on the day will decrease. There are reports that daily smoking, alcohol consumption, and exercise do not affect the efficacy of influenza vaccines [2]. If the general public is included in the target population, the degree of impact of lifestyle on the day of vaccination is unknown, but it may not change much.

Other factors, such as using analgesics or other medications, might have influenced the adverse reaction and risk of contracting influenza; however, they were not explored in the current study.

## 5. Conclusions

Routine activities on the vaccination day, such as personal hygiene (bathing), smoking, alcohol consumption, caffeine intake, and exercise, did not influence the efficacy of the influenza vaccine. Among the influenza vaccine adverse reactions, only strenuous exercise affected the incidence of systemic adverse reactions. On vaccination day, no other lifestyle-related restrictions are necessary, barring strenuous physical exercise.

## Figures and Tables

**Table 1 vaccines-09-00753-t001:** Participants’ background characteristics during the 2018–2019 and 2019–2020 winter seasons.

Background Characteristics	Overall		Tamba		Toyooka		Shiso	
	*n* = 2731		*n* = 1126		*n* = 1127		*n* = 478	
	*n*	%	*n*	%	*n*	%	*n*	%
Sex								
Female	2069	76.18	826	73.49	887	79.55	356	74.63
Male	647	23.82	298	26.51	228	20.45	121	25.37
Unknown	15		2		12		1	
Age (years: mean, SD)	42.70	12.71	44.13	13.65	41.28	12.15	42.71	11.25
Unknown	50		18		22		10	
Pregnant (female only)								
No	1940	98.08	777	98.11	828	98.1	335	97.95
Yes	38	1.92	15	1.89	16	1.9	7	2.05
Unknown	91		34		43		14	
Cancer/malignant tumor								
No	2550	99.22	1031	99.42	1065	99.35	454	98.48
Yes	20	0.78	6	0.58	7	0.65	7	1.52
Unknown	161		89		55		17	
Immunological disease/diseases caused by steroid use								
No	2558	99.46	1033	99.61	1068	99.44	457	99.13
Yes	14	0.54	4	0.39	6	0.56	4	0.87
Unknown	159		89		53		17	
Renal disease								
No	2559	99.53	1033	99.61	1068	99.53	1068	99.53
Yes	12	0.47	4	0.39	5	0.47	5	0.47
Unknown	160		89		54		−595	
Diabetes								
No	2558	99.49	1030	99.32	1069	99.72	459	99.35
Yes	13	0.51	7	0.68	3	0.28	3	0.65
Unknown	160		89		55		16	
Hepatic disease								
No	2562	99.69	1032	99.52	1070	99.81	460	99.78
Yes	8	0.31	5	0.48	2	0.19	1	0.22
Unknown	161		89		55		17	
Pulmonary disease								
No	2542	98.8	1026	98.65	1060	98.88	456	98.92
Yes	31	1.2	14	1.35	12	1.12	5	1.08
Unknown	158		86		55		17	
Body temperature at the time of vaccination (°C: mean, SD)	36.39	0.36	36.40	0.34	36.39	0.37	36.34	0.36
Unknown	151		66		60		25	
Physical condition at the time of vaccination								
Good	2605	98.15	1064	98.61	1077	97.64	464	98.31
Poor	49	1.85	15	1.39	26	2.36	8	1.69
Unknown	77		47		24		6	
History of influenza vaccination								
No	127	4.68	62	5.55	40	3.57	25	5.26
Yes	2584	95.32	1055	94.45	1079	96.43	450	94.74
Unknown	20		9		8		3	
Food/drug allergies								
No	2380	88.12	994	89.87	968	86.43	418	88
Yes	321	11.88	112	10.13	152	13.57	57	12
Unknown	30		20		7		3	
Personal hygiene status after vaccination on vaccination day								
Bath	1882	70.57	739	66.76	760	69.79	383	81.32
Shower only	699	26.21	317	28.64	308	28.28	74	15.71
Wiping	30	1.12	20	1.81	4	0.37	6	1.27
No washing	56	2.1	31	2.8	17	1.56	8	1.7
Unknown	64		19		38		7	
Alcohol consumption after vaccination on vaccination day								
No	2245	83.03	935	83.56	939	84.52	371	78.27
Yes	459	16.97	184	16.44	172	15.48	103	21.73
Unknown	27		7		16		4	
Caffeine intake after vaccination on vaccination day								
No	1312	49.83	606	55.29	508	47.43	198	42.49
Yes	1321	50.17	490	44.71	563	52.57	268	57.51
Unknown	98		30		56		12	
Smoking after vaccination on vaccination day								
Nonsmoker	2231	82.23	924	82.43	938	84.13	369	77.36
Previous smoker	299	11.02	134	11.95	96	8.61	69	14.47
Smoker, no smoking on the day of vaccination	11	0.41	9	0.8	2	0.18	0	0
Smoker, smoked on vaccination day	172	6.34	54	4.82	79	7.09	39	8.18
Unknown	18		5		12		1	
Exercise after vaccination on vaccination day								
No	2574	95.05	1077	96.07	1060	95.5	437	91.61
Yes	134	4.95	44	3.93	50	4.5	40	8.39
Unknown	23		5		17		1	
Contracted influenza in 2018–2020 or had an adverse reaction								
Contracted influenza								
No	2585	94.65	1086	96.45	1061	94.14	438	91.63
Yes	146	5.35	40	3.55	66	5.86	40	8.37
Unknown	0		0		0		0	
Type of influenza contracted								
A	114	78.08	36	90	53	80.3	25	62.5
B	15	10.27	2	5	13	19.7	0	0
A + B	1	0.68	1	2.5	0	0	0	0
Unknown	16	10.96	1	2.5	0	0	15	37.5
Redness at the injection site								
No	1001	36.95	554	49.38	281	25.2	166	35.17
Yes	1708	63.05	568	50.62	834	74.8	306	64.83
Unknown	22		4		12		6	
Swelling at the injection site								
No	1020	37.71	529	47.11	320	28.83	171	36.23
Yes	1685	62.29	594	52.89	790	71.17	301	63.77
Unknown	26		3		17		6	
Induration at the injection site								
No	1882	70.07	808	72.6	752	68.24	322	68.37
Yes	804	29.93	305	27.4	350	31.76	149	31.63
Unknown	45		13		25		7	
Pain at the injection site								
No	1413	52.31	647	57.66	518	46.92	248	52.21
Yes	1288	47.69	475	42.34	586	53.08	227	47.79
Unknown	30		4		23		3	
Heat sensation at the injection site								
No	1329	48.97	658	58.75	442	39.5	229	48.21
Yes	1385	51.03	462	41.25	677	60.5	246	51.79
Unknown	17		6		8		3	
Itching at the injection site								
No	1447	53.45	734	65.65	474	42.4	239	50.74
Yes	1260	46.55	384	34.35	644	57.6	232	49.26
Unknown	24		8		9		7	
Heaviness/lassitude at the injection site								
No	2145	79.77	910	81.69	860	77.62	375	80.3
Yes	544	20.23	204	18.31	248	22.38	92	19.7
Unknown	42		12		19		11	
Other localized symptoms								
No	2470	98.33	1039	98.58	983	98.3	448	97.82
Yes	42	1.67	15	1.42	17	1.7	10	2.18
Unknown	219		72		127		20	
Some types of localized symptoms								
No	491	18.33	280	25.34	124	11.25	87	18.43
Yes	2188	81.67	825	74.66	978	88.75	385	81.57
Unknown	52		21		25		6	
Fever								
No	2672	98.34	1104	98.13	1098	98.04	470	99.58
Yes	45	1.66	21	1.87	22	1.96	2	0.42
Unknown	14		1		7		6	
Chills								
No	2673	98.34	1105	98.22	1102	98.39	466	98.52
Yes	45	1.66	20	1.78	18	1.61	7	1.48
Unknown	13		1		7		5	
Headache								
No	2588	95.11	1076	95.64	1059	94.39	453	95.57
Yes	133	4.89	49	4.36	63	5.61	21	4.43
Unknown	10		1		5		4	
Fatigue								
No	2486	91.33	1036	92.09	1022	91.09	428	90.11
Yes	236	8.67	89	7.91	100	8.91	47	9.89
Unknown	9		1		5		3	
Nasal discharge								
No	2618	96.21	1093	97.07	1073	95.72	452	95.36
Yes	103	3.79	33	2.93	48	4.28	22	4.64
Unknown	10		0		6		4	
Cough								
No	2658	97.61	1101	97.78	1090	97.06	467	98.52
Yes	65	2.39	25	2.22	33	2.94	7	1.48
Unknown	8		0		4		4	
Nausea								
No	2698	99.19	1114	99.02	1116	99.55	468	98.73
Yes	22	0.81	11	0.98	5	0.45	6	1.27
Unknown	11		1		6		4	
Diarrhea								
No	2682	98.64	1105	98.22	1107	98.75	470	99.37
Yes	37	1.36	20	1.78	14	1.25	3	0.63
Unknown	12		1		6		5	
Difficulty moving the upper limbs								
No	2523	92.86	1047	93.15	1032	92.06	444	94.07
Yes	194	7.14	77	6.85	89	7.94	28	5.93
Unknown	14		2		6		6	
Numbness								
No	2677	98.42	1107	98.49	1103	98.31	467	98.52
Yes	43	1.58	17	1.51	19	1.69	7	1.48
Unknown	11		2		5		4	
Other systemic symptoms								
No	2593	98.41	1071	97.81	1054	98.6	468	99.36
Yes	42	1.59	24	2.19	15	1.4	3	0.64
Unknown	96		31		58		7	
Some types of systemic symptoms								
No	2129	80.01	887	80.49	868	79.63	374	79.74
Yes	532	19.99	215	19.51	222	20.37	95	20.26
Unknown	70		24		37		9	
Some types of systemic or localized symptoms								
No	465	17.34	263	23.76	118	10.71	84	17.8
Yes	2216	82.66	844	76.24	984	89.29	388	82.2
Unknown	50		19		25		6	

Tamba: Hyogo Prefectural Tamba Medical Center, Toyooka: Toyooka Public Hospital, Shiso: Shiso Municipal Hospital, SD: standard deviation.

**Table 2 vaccines-09-00753-t002:** Effect of lifestyle factors on the incidence of influenza.

Lifestyle Factors	Contracted Influenza		
	RR (95% CI)		
	Model 1	Model 2	Model 3
Personal hygiene			
Bathing	1.09 (0.76–1.56)	1.25 (0.85–1.83)	1.20 (0.80–1.79)
Smoking			
Previous smoker	1.22 (0.76–1.96)	1.33 (0.81–2.21)	1.35 (0.79–2.30)
Smoker, did not smoke on the vaccination day	Not estimated	Not estimated	Not estimated
Smoker, smoked on the vaccination day	1.12 (0.60–2.09)	1.13 (0.58–2.21)	1.17 (0.58–2.36)
Alcohol consumption			
Yes	1.03 (0.68–1.56)	1.12 (0.72–1.72)	1.15 (0.73–1.80)
Caffeine intake			
Yes	0.81 (0.58–1.12)	0.89 (0.63–1.25)	0.88 (0.62–1.25)
Exercise			
Yes	0.83 (0.37–1.84)	0.71 (0.30–1.71)	0.78 (0.33–1.86)

Model 1. Rough analysis; Model 2. Adjusted for sex/age/underlying conditions; Model 3. Adjusted for sex/age/underlying conditions/lifestyle, CI, confidence interval; RR, risk ratio.

**Table 3 vaccines-09-00753-t003:** Effect of lifestyle factors on the incidence of some types of adverse reactions (local or systemic).

Lifestyle Factors	Some Types of Adverse Reactions Observed		
	RR (95% CI)		
	Model 1	Model 2	Model 3
Personal hygiene			
Bathing	1.00 (0.96–1.04)	0.99 (0.96–1.03)	1.00 (0.96–1.04)
Smoking			
Previous smoker	0.91 (0.85–0.97)	1.00 (0.94–1.07)	1.00 (0.93–1.07)
Smoker, did not smoke on vaccination day	0.76 (0.48–1.18)	0.82 (0.52–1.29)	0.77 (0.45–1.34)
Smoker, smoked on vaccination day	0.90 (0.82–0.98)	1.01 (0.92–1.10)	1.00 (0.91–1.10)
Alcohol consumption			
Yes	0.93 (0.89–0.98)	1.01 (0.96–1.06)	1.00 (0.95–1.06)
Caffeine intake			
Yes	1.01 (0.97–1.04)	1.02 (0.99–1.06)	1.02 (0.98–1.06)
Exercise			
Yes	0.96 (0.88–1.05)	1.05 (0.97–1.14)	1.06 (0.98–1.15)

Model 1. Rough analysis; Model 2. Sex/age/underlying conditions adjusted; Model 3. Sex/age/underlying conditions/lifestyle adjusted, CI, confidence interval; RR, risk ratio.

**Table 4 vaccines-09-00753-t004:** Effect of lifestyle factors on local adverse reaction incidence.

Lifestyle Factors	Incidence of Local Adverse Reactions		
	RR (95% CI)		
	Model 1	Model 2	Model 3
Personal hygiene			
Bathing	1.00 (0.96–1.04)	0.99 (0.96–1.04)	1.00 (0.96–1.04)
Smoking			
Previous smoker	0.91 (0.85–0.97)	1.01 (0.95–1.09)	1.01 (0.94–1.08)
Smoker, did not smoke on the vaccination day	0.76 (0.49–1.19)	0.84 (0.54–1.31)	0.79 (0.46–1.37)
Smoker, smoked on the vaccination day	0.88 (0.81–0.97)	1.00 (0.91–1.10)	0.99 (0.90–1.09)
Alcohol consumption			
Yes	0.92 (0.87–0.98)	1.00 (0.95–1.06)	1.00 (0.95–1.06)
Caffeine intake			
Yes	1.02 (0.98–1.05)	1.03 (1.00–1.07)	1.03 (0.99–1.07)
Exercise			
Yes	0.95 (0.86–1.04)	1.04 (0.95–1.13)	1.06 (0.97–1.15)

Model 1. Rough analysis; Model 2. Adjusted for sex/age/underlying conditions; Model 3. Adjusted for sex/age/underlying conditions/lifestyle, CI, confidence interval; RR, risk ratio.

**Table 5 vaccines-09-00753-t005:** Effect of lifestyle factors on systemic adverse reaction incidence.

Lifestyle Factors	Incidence of Systemic Adverse Reactions		
	RR (95% CI)		
	Model 1	Model 2	Model 3
Personal hygiene			
Bathing	1.01 (0.85–1.19)	1.03 (0.86–1.24)	1.04 (0.87–1.26)
Smoking			
Previous smoker	0.89 (0.68–1.15)	0.99 (0.75–1.31)	0.98 (0.73–1.31)
Smoker, did not smoke on the vaccination day	0.44 (0.07–2.87)	0.72 (0.11–4.75)	Not estimated
Smoker, smoked on the vaccination day	0.83 (0.59–1.17)	1.00 (0.69–1.44)	0.88 (0.58–1.33)
Alcohol consumption			
Yes	0.97 (0.79–1.19)	1.07 (0.86–1.34)	1.07 (0.85–1.34)
Caffeine intake			
Yes	0.99 (0.85–1.15)	1.00 (0.85–1.18)	0.99 (0.84–1.17)
Exercise			
Yes	1.20 (0.88–1.65)	1.35 (0.99–1.85)	1.44 (1.06–1.97)

Model 1. Rough analysis; Model 2. Adjusted for sex/age/underlying conditions; Model 3. Adjusted for sex/age/underlying conditions/lifestyle, CI, confidence interval; RR, risk ratio.

## Data Availability

The data sets used and/or analyzed during the present study are available from the first author on reasonable request.

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
