# Peer review of "Effects of Vaccination Day Routine Activities on Influenza Vaccine Efficacy and Vaccination-Induced Adverse Reaction Incidence: A Cohort Study"

_vaccines, 2021, doi:10.3390/vaccines9070753_

Round 1
Reviewer 1 Report
The authors investigate an important aspect of vaccination by studying the effect of vaccination on daily activity and adverse events.
Comments:
1) The introduction is extremely brief, please add additional background information.
a) Basic biology of influenza is needed
b) Please expand/explain the use of bath houses and the personal hygiene, smoking, alcohol consumption, or exercising, on the efficacy and adverse re-actions of the influenza vaccine.
c) Why were the personal hygiene smoking, alcohol consumption, or exercising, on the efficacy and adverse actions of the influenza vaccine investigated.
2) Similarly the materials and methods section needs additional information. Few examples below.
a) 2.1 how were the participants recruited and enrolled?
b) 2.2.3 what clinical assay was used to confirm the incidence of influenza.
c) 2.3 is comprised of a single sentence. How was the data analyzed?
3) Edits are required throughout the manuscript.
few examples.
a) Line 49, the word "changing" is in different font and size.
b) line 49, the use of "catch a cold" is inappropriate. An individual can be infect with influenza.
c) Line 95, not sure what "Vaccine screening questionnaires and questionnaires were" means.
Author Response
Reviewer 1’s Comments and Suggestions for Authors:
The authors investigate an important aspect of vaccination by studying the effect of vaccination on daily activity and adverse events.
Comments:
1) The introduction is extremely brief, please add additional background information.
- a) Basic biology of influenza is needed
Response: Thank you for the valuable comments. Accordingly, we have added a description on the basic biology of influenza and influenza vaccine to the introduction section of the revised manuscript. (Pages 1,2 Lines 43-51, 54-58)
- b) Please expand/explain the use of bath houses and the personal hygiene, smoking, alcohol consumption, or exercising, on the efficacy and adverse re-actions of the influenza vaccine.
Response: Thank you for the valuable comment. Accordingly, we have added explanations on the use of bath houses, personal hygiene, daily smoking, alcohol consumption, and exercising, on the efficacy and adverse re-actions of the influenza vaccine to the introduction section of the revised manuscript. (Page 2, lines 64-67, 70-73)
- c) Why were the personal hygiene smoking, alcohol consumption, or exercising, on the efficacy and adverse actions of the influenza vaccine investigated.
Response: In Japan, healthcare professionals document details about routine activities, of each patient, performed on the day of vaccination, such as personal hygiene (bathing), smoking, alcohol consumption, and exercising, for accessing the efficacy and adverse reactions of the influenza vaccine.
2) Similarly the materials and methods section needs additional information. Few examples below.
- a) 2.1 how were the participants recruited and enrolled?
Response: Thank you for the valuable comments. In Japan, a mass Influenza vaccination program is conducted, every year before the influenza season, for healthcare professionals in the hospital. All health care professionals who were vaccinated against influenza vaccine were eligible to be included in our study. We have added this information to the Participants sub-section in the methods section of the revised manuscript. (Page 2, Lines 90-93)
- b) 2.2.3 what clinical assay was used to confirm the incidence of influenza.
Response: Influenza was diagnosed using rapid influenza diagnostic tests. We have added this information to the Incidence of Influenza sub-section in the methods section of the revised manuscript. (Page 3, Lines 121-124)
- c) 2.3 is comprised of a single sentence. How was the data analyzed?
Response: We have revised the sub-heading “Data Analysis” to “ Data Analysis of Participants’ Background Characteristics”. The results of participant characteristics were simply calculated. (Page 3, Lines 136)
3) Edits are required throughout the manuscript.
few examples.
- Line 49, the word "changing" is in different font and size.
Response: Thank you for pointing this out. We have revised the indicated text accordingly. (Page 2, Line 64)
- line 49, the use of "catch a cold" is inappropriate. An individual can be infect with influenza.
Response: We apologize for the inappropriate use of the phrase. By “catch a cold” we only intended to mean “get sick,” not necessarily contract influenza. We have now revised the sentence as follows: “which may cause people to get sick,” to improve clarity. (Page 2, Line 65)
- Line 95, not sure what "Vaccine screening questionnaires and questionnaires were" means.
Response: We apologize for the ambiguity in the text. We have revised the text as follows: “Each participant was provided a vaccine screening healthcare questionnaire prior to vaccine administration. This is a standard procedure in Japan, and the standardized questionnaire was analyzed in this study” to improve clarity. (Pages 2,3 Lines 95-96)
Reviewer 2 Report
This is an excellent piece of work of general interest and has been well written. My only question to the authors as mentioned in the limitation section would be why were the healthcare professional only selected for this study? Will including the general public change the observations significantly.
Author Response
This is an excellent piece of work of general interest and has been well written. My only question to the authors as mentioned in the limitation section would be why were the healthcare professional only selected for this study? Will including the general public change the observations significantly.
Response: Thank you for the valuable comments. Accordingly, we have now explained the reason for selecting only healthcare professional in the study, in the limitations sub-section of the revised manuscript. (Page 13, Lines 271-274)
Round 2
Reviewer 1 Report
The authors have addressed most of the comments. However, please address the comment below.
Section 2.7, what does "The results of participant characteristics were simply calculated." mean? Please state what was done with the data.
Author Response
Reviewer 1’s Comments and Suggestions for Authors:
The authors have addressed most of the comments. However, please address the comment below.
Comments: Section 2.7, what does " The results of participant characteristicswere simply calculated." mean? Please state what was done with the data.
Response: We apologize for the inappropriate use of the phrase. We have now revised the sentence as follows: “Means of continuous variables and percentiles of categorical variables were calcu-lated to characterize participants.,” to improve clarity. (Page 3, Lines 136-137)